# CausalAF: Causal Autoregressive Flow for Safety-Critical Driving Scenario Generation

**Wenhao Ding[1], Haohong Lin[1], Bo Li[2], Ding Zhao[1]**
[1]Carnegie Mellon University
[2]University of Illinois at Urbana-Champaign
{wenhaod, haohongl}@andrew.cmu.edu, lbo@illinois.edu, dingzhao@cmu.edu

**Abstract:** Generating safety-critical scenarios, which are crucial yet difficult to collect, provides an effective way to evaluate the robustness of autonomous driving systems. However, the diversity of scenarios and efficiency of generation methods are heavily restricted by the rareness and structure of safety-critical scenarios. Therefore, existing generative models that only estimate distributions from observational data are not satisfying to solve this problem. In this paper, we integrate causality as a prior into the scenario generation and propose a flow-based generative framework, *Causal Autoregressive Flow (CausalAF)*. *CausalAF* encourages the generative model to uncover and follow the causal relationship among generated objects via novel causal masking operations instead of searching the sample only from observational data. By learning the cause-and-effect mechanism of how the generated scenario causes risk situations rather than just learning correlations from data, *CausalAF* significantly improves learning efficiency. Extensive experiments on three heterogeneous traffic scenarios illustrate that *CausalAF* requires much fewer optimization resources to effectively generate safety-critical scenarios. We also show that using generated scenarios as additional training samples empirically improves the robustness of autonomous driving algorithms.

**Keywords:** Causal Generative Models, Scenario Generation, Autonomous Driving

## 1 Introduction

According to a recent report [1], several companies have made their autonomous vehicles (AVs) drive more than 10,000 miles without disengagement. It seems that current AVs have achieved great success in normal scenarios that cover most cases in daily life. However, we are still unsure about their performance under critical cases, which could be too rare to collect in the real world. For example, a kid suddenly running into the drive lane chasing a ball leaves the AV a very short time to react. This kind of situation, named *safety-critical scenarios*, could be the last puzzle to evaluate the safety of AVs before deployment.

Generating safety-critical scenarios with Deep Generative Models (DGMs), which estimate the distribution of data samples with neural networks, is viewed as a promising way in recent works [2]. Existing literature either searches in the latent space to build scenarios [3, 4] or directly uses optimization to find the adversarial examples [5, 6]. However, such a generation task is still challenging since we are required to simultaneously consider fidelity to avoid conjectural scenarios that will never happen in the real world, as well as the safety-critical level which is indeed rare compared with normal scenarios. In addition, generating reasonable threats to vehicles' safety can be inefficient if the model purely relies on unstructured observational data, as the safety-critical scenarios are rare and follow fundamental physical principles. Inspired by the fact that humans are good at abstracting the causation beneath the observations with prior knowledge, we explore a new direction toward causal generative models for this generation task.

6th Conference on Robot Learning (CoRL 2022), Auckland, New Zealand.

To have a glance at causality in traffic scenarios, we show an example in Figure 1(b). When a vehicle $B$ is parked in the middle between the autonomous vehicle $A$ and pedestrian $C$, the view of $A$ is blocked, making $A$ have little time to brake and thus have a potential collision with $C$. As human drivers, we believe $B$ should be the cause of the accident. This scenario may take AVs millions of hours to collect [7]. Even if we use traditional generative models to generate this scenario, the model tends to memorize the location of all objects without learning the reasons. As a remedy, we can incorporate causality into generative models for the efficient generation of such safety-critical scenarios.

In this paper, we propose a structured generative model with causal priors. We model the causality as a directed acyclic graph (DAG) named Causal Graph (CG) [8]. To facilitate CG in the traffic scenario, we propose another Behavioral Graph (BG) for representing the interaction between objects in scenarios. The graphical representation of both graphs makes it possible to use the BG to unearth the causality given by CG. Based on BG, we propose the first generative model that integrates causality into the graph generation task and names it *CausalAF*. Specifically, we propose two types of causal masks – Causal Order Masks (COM) that modifies the node order for node generation, and Causal Visibility masks (CVM) that removes irrelevant information for edge generation. We show the diagram of *CausalAF* generation in Figure 1(a) and summarize our main contributions as following:

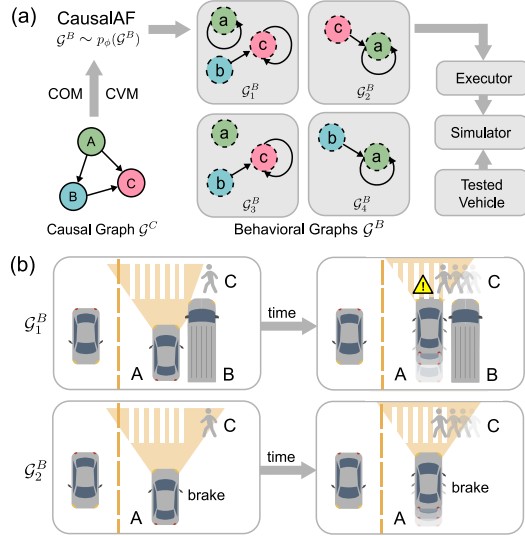

Figure 1: **(a)** Diagram of the generation pipeline using *CausalAF*. **(b)** Two scenarios obtained by two Behavioral Graphs shows the causality behind scenarios. The top one is safety-critical because the view of vehicle $A$ is blocked by vehicle $B$.

- We propose a causal generative model *CausalAF* that integrates causal graphs with two novel mask operators for safety-critical scenario generation.
- We show that *CausalAF* dramatically improves the efficiency and performance on three standard traffic settings compared with purely data-driven baselines.
- We show that the training on generated safety-critical scenarios improves the robustness of 4 reinforcement learning-based driving algorithms.

## 2 Graphical Representation of Scenarios

We start by proposing a novel representation of traffic scenarios using a graph structure. Then, we propose to generate such a graphical representation with an autoregressive generative model.

### 2.1 Behavioral Graph

Traffic scenarios mainly consist of interactions between static and dynamic objects, which can be naturally described by a graph structure. Therefore, we define Behavioral Graph $\mathcal{G}^B$ to represent driving scenarios with the following definition.

**Definition 1** (Behavioral Graph, BG). *Suppose a scenario has maximum $m$ objects with $n$ types. A Behavioral Graph $\mathcal{G}^B = (V^B, E^B)$ is a directed graph with node matrix $V^B \in \mathbb{R}^{m \times n}$ representing the types of objects and edge matrix $E^B \in \mathbb{R}^{m \times m \times (h_1 + h_2)}$ representing the interaction between objects, where $h_1$ is the number of edge types and $h_2$ is the dimension of edge attributes.*

According to this definition, $\mathcal{G}^B$ works as a planner that controls the behaviors of objects in the scenario based on the types of nodes $V^B$ and edges $E^B$. For example, two nodes $v_1$ and $v_2$ represent two vehicles and the edge from $v_1$ to $v_2$ represents the relative velocity from $v_1$ to $v_2$. Specifically, a self-loop edge $(i, i)$ represents that one object takes one action irrelevant to other objects (e.g., a

car goes straight or turns left with no impact on other road users), while other edges $(i, j)$ means object $i$ takes one action related to object $j$ (e.g., a car $i$ moves towards a pedestrian $j$). The edge attributes represent the properties of actions. For instance, the attribute $[x, y, v_x, v_y]$ of one edge has the following meaning: $x$ and $y$ are positions, and $v_x$ and $v_y$ are the velocities.

## 2.2 Behavioral Graph Generation with Autoregressive Flow

Generally, there are two ways to generate graphs: one is simultaneously generating all nodes and edges, and the other is iteratively generating nodes and adding edges between nodes. Considering the directed nature of $\mathcal{G}^B$, we utilize the Autoregressive Flow model (AF) [9], which is a type of sequentially DGMs, to generate nodes and edges of $\mathcal{G}^B$ step by step. It uses a invertible and differentiable transformation $\mathcal{F}_\phi$ parametrized by $\phi$ to convert the graph $\mathcal{G}^B$ to a latent variable $z$ that follows a base distribution $p(z)$ (e.g., Normal distribution $\mathcal{N}(0, I)$). According to the change of variables theorem, we can obtain $p_\phi(\mathcal{G}^B) = p(\mathcal{F}_\phi(\mathcal{G}^B)) \left| \det \frac{\partial \mathcal{F}_\phi(\mathcal{G}^B)}{\partial \mathcal{G}^B} \right|$. To increase the representing capability, $\mathcal{F}_\phi$ contains multiple functions $f_i$ for $i \in \{0, \dots, K\}$. The entire transformation is represented as $\mathcal{G}^B = z_K = f_K^{-1} \circ \cdots \circ f_0^{-1} \triangleq \mathcal{F}_\phi^{-1}(z_0)$ by repeatedly substituting the variable for the new variable $z_i$, where $\circ$ means the function composition. Eventually, we obtain the likelihood

$$\log p_\phi(\mathcal{G}^B) = p(z_0) - \sum_{i=1}^{K} \log \left| \det \frac{df_i^{-1}}{dz_{i-1}} \right|, \tag{1}$$

which will be used to learn the parameter $\phi$ based on empirical distribution of $\mathcal{G}^B$. After training, we can sample from $p_\phi(\mathcal{G}^B)$ by using the reverse function $\mathcal{F}_\phi^{-1}$. Let $V_{[i]}^B \in \mathbb{R}^n$ and $E_{[i,j]}^B \in \mathbb{R}^{h_1+h_2}$ represent node $i$ and edge $(i, j)$ of $\mathcal{G}^B$, then we can generate them with the sampling procedure:

$$V_{[i]}^B \sim \mathcal{N}\left(\mu_i^v, (\sigma_i^v)^2\right) = \mu_i^v + \sigma_i^v \odot \epsilon \text{ and } E_{[i,j]}^B \sim \mathcal{N}\left(\mu_{ij}^e, (\sigma_{ij}^e)^2\right) = \mu_{ij}^e + \sigma_{ij}^e \odot \epsilon, \tag{2}$$

where $\odot$ denotes the element-wise product and $\epsilon$ follows a Normal distribution $\mathcal{N}(0, I)$. Variables $\mu_i^v, \sigma_i^v, \mu_{ij}^e$, and $\sigma_{ij}^e$ are obtained from $\mathcal{F}_\phi$ in an autoregressive manner:

$$\mu_i^v, \sigma_i^v = \mathcal{F}_\phi\left(V_{[0:i-1]}^B, E_{[0:i-1,0:m]}^B\right) \text{ and } \mu_{ij}^e, \sigma_{ij}^e = \mathcal{F}_\phi\left(V_{[0:i]}^B, E_{[0:i,0:j-1]}^B\right), \tag{3}$$

where $[0 : i]$ represents the elements from index 0 to index $i$. After the sampling, we obtain the node and edge type by converting $V^B$ and part of $E_B$ from continuous values to one-hot vectors:

$$V_{[i]}^B \leftarrow \text{onehot}\left[\arg\max(V_{[i]}^B)\right], \quad E_{[i,j,0:h_1]}^B \leftarrow \text{onehot}\left[\arg\max(E_{[i,j,0:h_1]}^B)\right] \quad \forall i, j \in [m]. \tag{4}$$

Intuitively, the generation of one node depends on all previously generated nodes and edges. One node only has edges pointing to the nodes that are generated before it. To illustrate this autoregressive generation process, we provide an example with three nodes in Figure 2(a).

# 3 Causal Autoregressive Flow (CausalAF)

In this section, we discuss how to integrate causality into the autoregressive generating process of the Behavioral Graph $\mathcal{G}^B$. In general, we transfer the prior knowledge from a causal graph to $\mathcal{G}^B$ by increasing the structural similarity. However, calculating such similarity is not easy because of the discrete nature of graphs. To solve this problem, we propose *CausalAF* with two causal masks, i.e., Causal Order Masks (COM) and Causal Visible Masks (CVM), that make the generated $\mathcal{G}^B$ follow the causal information.

## 3.1 Causal Generative Models

**Definition 2** (Structural Causal Models [10], SCM). *A structural causal model (SCM) $\mathfrak{C} := (S, U)$ consists of a collection $S$ of $m$ functions, $X_j := f_j(PA_j, U_j), \forall j \in [m]$, where $PA_j \subset \{X_1, \dots, X_m\} \backslash \{X_j\}$ are called parents of $X_j$; and a joint distribution $U = \{U_1, \dots, U_m\}$ over the noise variables, which are required to be jointly independent.*

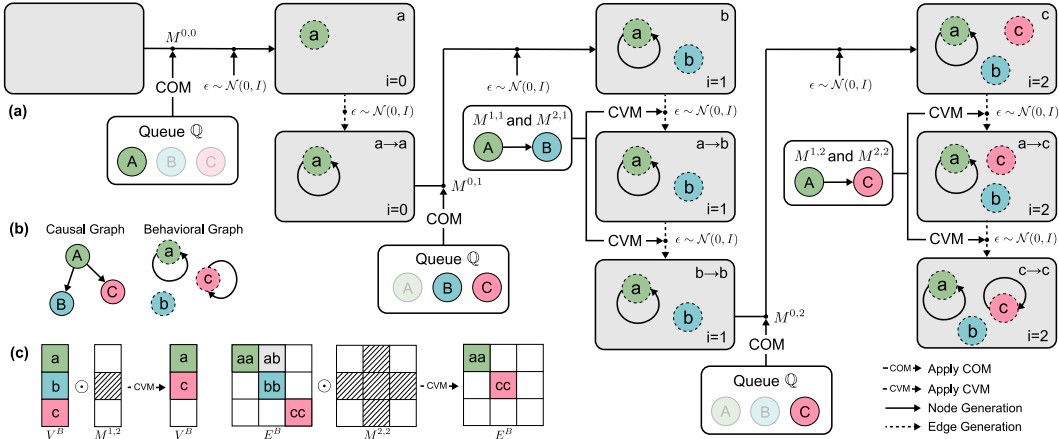

Figure 2: **(a)** The generation process of a BG, which starts from an empty graph. We add one node or one edge at each step. COM is applied to select nodes following the CG and CVM is applied to mask out non-parent nodes following the CG. **(b)** CG and BG used in the example. **(c)** The explanation of CVM when generating edges for $c$, where irrelevant node $b$ is masked out in both $V^B$ and $E^B$.

**Definition 3** (Causal Graphs [10], CG). *The causal graph $\mathcal{G}^C$ of an SCM is obtained by creating one node for each $X_j$ and drawing directed edges from each parent in $\boldsymbol{PA}_j(\mathcal{G}^C)$ to $X_j$. The representation of $\mathcal{G}^C = (V^C, E^C)$ consists of the node vector $V^C \in \{0,1\}^m$ and the adjacency matrix $E^C \in \{0,1\}^{m \times m \times h_1}$. Each edge $(i,j)$ represents a causal relation from node $i$ to node $j$.*

We formally describe the causality based on the above definitions of SCM and CG. In fact, the generative model $p_\phi(\mathcal{G}^B)$ mentioned in Section 2 shares a very similar definition with SCM except that $\mathcal{G}^B$ does not follow the order of causality. This inspires us that we can convert $p_\phi(\mathcal{G}^B)$ to an SCM by incorporating the causal graph $\mathcal{G}^C$ into the generation process. In this paper, we assume the causal graph $\mathcal{G}^C$ can be summarized by expert knowledge. Therefore, we incorporate a given $\mathcal{G}^C$ into $p_\phi(\mathcal{G}^B|\mathcal{G}^C)$ by regularizing the generative process with two novel masks as shown in Figure 2.

### 3.2 Causal Graph Integration

**Causal Order Masks (COM)** The order is vital during the generation of $\mathcal{G}^B$ since we must ensure the cause is generated before the effect. To achieve this, we maintain a priority queue $\mathbb{Q}$ to store the valid child types according to the causal relation in $\mathcal{G}^C$. $\mathbb{Q}$ is initialized with $\mathbb{Q} = \{i | \boldsymbol{PA}_i(\mathcal{G}^C) = \emptyset, \forall i \in [m]\}$, which contains all nodes that do not have parent nodes. Then, in each node generation step, we update $\mathbb{Q}$ by removing the generated node $i$ and adding the child nodes of $i$. Since one node may have multiple parents thus it is valid only if all of its parents have been generated. We use $\mathbb{Q}$ to create a $k$-hot mask $M^{0,i} \in \mathbb{R}^n$, where the element is set to 1 if it is a valid type. Then, we apply COM to the node matrix by $V^B_{[i]} \leftarrow M^{0,i} \odot V^B_{[i]}$, where $V^B_{[i]}$ is the node vector obtained from $\mathcal{F}_\phi$ for node $i$. Intuitively, this mask sets the probability of the invalid node types to 0 to make sure the generated node always follows the correct order.

**Causal Visible Masks (CVM)** Ensuring a correct causal order is still insufficient to represent the causality. Thus, we further propose another type of mask called CVM, which removes the non-causal connections, i.e., non-parent nodes to the current node in $\mathcal{G}^C$, when generating edges. Specifically, we generate two binary masks $M^{1,i} \in \mathbb{R}^{m \times n}$ and $M^{2,i} \in \mathbb{R}^{m \times m \times (h_1+h_2)}$ with $M^{1,i}_{[j,:]} = 0$ and $M^{2,i}_{[j,i,:]} = 0, \forall j \notin \boldsymbol{PA}_i(\mathcal{G}^C)$. Then, we apply them to update the node matrix and edge matrix by $V^B \leftarrow M^{1,i} \odot V^B$ and $E^B \leftarrow M^{2,i} \odot E^B$. We illustrate an example of this process in Figure 2(c). Assume we are generating edges for node $c$. We need to remove node $b$ since $\mathcal{G}^C$ tells us that $B$ does not have edges to node $C$. After applying $M^v$ and $M^e$, we move the features of node $c$ to the previous position of $b$. This permuting operation is important since the autoregressive model is not permutation invariant.

### 3.3 Optimization of Safety-critical Generation

After introducing the generative process of *CausalAF*, we now turn to the optimization procedure. The target is to generate scenarios $\tau = \mathcal{E}(\mathcal{G}^B)$ with an executor $\mathcal{E}$ to satisfy a given goal, which is formulated as an objective function $\mathcal{L}_g$. We define $\mathcal{L}_g(\tau) = \mathbb{1}(D(\tau) < \epsilon)$, where $D(\tau)$ represents the minimal distance between the autonomous vehicle and other objects and $\epsilon$ is a small threshold. Therefore, the optimization is to solve the problem $\max_\phi \mathbb{E}_{\mathcal{G}^B \sim p_\phi(\mathcal{G}^B|\mathcal{G}^C)}[\mathcal{L}_g(\mathcal{E}(\mathcal{G}^B))]$. Usually, $\mathcal{L}_g$ contains non-differentiable operators (e.g., complicated simulation and rendering), thus we have to utilize black-box optimization methods to solve the problem. We consider a policy gradient algorithm named REINFORCE [11], which obtains the estimation of the gradient from samples by

$$\nabla_\phi \mathbb{E}_{\mathcal{G}^B \sim p_\phi(\mathcal{G}^B|\mathcal{G}^C)}[\mathcal{L}_g(\mathcal{E}(\mathcal{G}^B))] = \mathbb{E}[\nabla_\phi \log p(\mathcal{G}^B|\mathcal{G}^C)\mathcal{L}_g(\mathcal{E}(\mathcal{G}^B))] \qquad (5)$$

Overall, the entire training algorithm is summarized in **Algorithm** 1. In addition, we can prove that the *CausalAF* guarantees monotonicity of likelihood in Theorem 1 at convergence. The detail of the proof is given in Appendix A.

**Theorem 1** (Monotonicity of Likelihood). *Given the true causal graph $\mathcal{G}^{C^*} = (V^C, E^{C^*})$ and distance SHD [12], for CG $\mathcal{G}_1^C = (V^C, E_1^C)$ and $\mathcal{G}_2^C = (V^C, E_2^C)$, if $SHD(\mathcal{G}_1^C, \mathcal{G}^{C^*}) < SHD(\mathcal{G}_2^C, \mathcal{G}^{C^*})$, and $\exists\, e$, s.t. $E_1^C \cup \{e\} = E_2^C$, CausalAF converges with the monotonicity of likelihood for collision samples, i.e. $p_\phi(D(\tau) < \epsilon \mid \mathcal{G}_2^C) < p_\phi(D(\tau) < \epsilon \mid \mathcal{G}_1^C) < p_\phi(D(\tau) < \epsilon \mid \mathcal{G}^{C^*})$.*

### 3.4 Scenario Sampling and Execution

Thanks to the autoregressive generation of CausalAF, we are able to conduct generation conditioned on arbitrary numbers or types of nodes. Instead of generating from the scratch, we can start from an existing $\mathcal{G}_c^B$ for the generation with $\mathcal{G}^B \sim p_\phi(\cdot|\mathcal{G}_c^B, \mathcal{G}^C)$. The conditional generation can be used for interactive scenarios, e.g., using the autonomous vehicle's information or the data of partial scenarios in the real world as conditions to generate diverse and realistic scenarios. After sampling the scenarios, the physical properties (e.g., position and velocity) defined in the generated $\mathcal{G}^B$ are executed in the simulator $\mathcal{E}$ to create sequential scenarios $\tau$. After the execution, the simulator outputs the objective function $L_g(\tau)$ as the result.

---

**Algorithm 1:** Training process of CausalAF

**Input:** Causal Graph $\mathcal{G}^C$, Goal $\mathcal{L}_g$, Learning rate $\alpha$, Maximum node number $m$

**while** $\phi$ *not converged* **do**
  // Sample a BG $\mathcal{G}^B \sim p_\phi(\mathcal{G}^B|\mathcal{G}^C)$
  **for** $i < m$ **do**
    Sample node matrix $V_{[i]}^B$ by (2)
    Get node type $V_{[i]}^B$ by (4)
    Apply COM $M^{0,i}$ to $V_{[i]}^B$
    Apply CVM $M^{1,i}, M^{2,i}$ to $V_{[i]}^B, E_{[i,j]}^B$
    **for** $j \leq i$ **do**
      Sample edge matrix $E_{[i,j]}^B$ by (2)
      Get edge type $E_{[i,j]}^B$ by (4)
  Collect one scenario $\mathcal{G}^B = \{V^B, E^B\}$
  // Learn model parameters
  Calculate the likelihood $p_\phi(\mathcal{G}^B|\mathcal{G}^C)$
  Execute $\tau = \mathcal{E}(\mathcal{G}^B)$ and get $\mathcal{L}_g(\tau)$
  Use (5) to update $\phi \leftarrow \phi - \alpha \nabla_\phi \mathcal{L}_g(\tau)$

---

## 4 Experiment

We evaluate *CausalAF* using three top pre-crash traffic scenarios defined by U.S. Department of Transportation [13] and Euro New Car Assessment Program [14]. Our empirical results show that it may not be trivial for the generative models to learn the underlying causality even if such causality seems understandable to humans. Particularly, we conduct a series of experiments to answer the following main questions: **Q1**: How does CausalAF perform compared to other scenario generate methods? **Q2**: How does causality help the generation process? **Q3**: How can we use the generated safety-critical scenarios? In this section, we will first introduce the designed environment and baseline methods. Then we will answer the above questions by carefully investigating the experiment results.

### 4.1 Experiment Design and Setting

**Scenario.** We consider three safety-critical traffic scenarios (shown in Figure 3) that have clear causation. The causal graph $\mathcal{G}^C$ for each scenario is displayed on the upper right of the scenario.

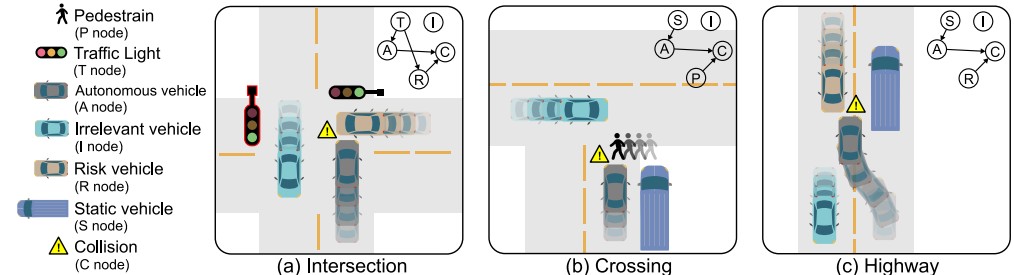

Figure 3: Three causal traffic scenarios are used in our experiments. The corresponding causal graphs are shown in the upper right of each scenario. Please refer to Section 4.1 for details.

Table 1: Collision rate (↑) of generated safety-critical scenarios. **Bold** font means the best.

| Method | L2C [5] | MMG [4] | SAC [15] | STRIVE [16] | Baseline | Baseline+COM | CausalAF |
|---|---|---|---|---|---|---|---|
| Intersection | 0.63±0.28 | 0.31±0.54 | 0.47±0.61 | 0.64±0.12 | 0.29±0.84 | 0.69±0.52 | **0.98±0.01** |
| Crossing | 0.69±0.41 | 0.43±0.56 | 0.38±0.49 | 0.55±0.10 | 0.35±0.65 | 0.57±0.48 | **0.83±0.13** |
| Highway | 0.85±0.10 | 0.56±0.36 | 0.58±0.41 | 0.67±0.16 | 0.53±0.69 | 0.88±0.04 | **0.91±0.06** |

- **Intersection**. One potential safety-critical event could happen when the traffic light $T$ turns from green to yellow to give the road right to an autonomous vehicle $A$. Here, $A$ and $R$ are influenced by $T$. $R$ runs the red light, colliding with $A$ perpendicularly, therefore, causing the collision $C$ together with $A$. $I$ does not influence other objects.

- **Crossing**. A pedestrian $P$ and an autonomous vehicle $A$ are crossing the road in vertical directions. There also exists a static vehicle $S$ parked by the side of the road. Then a potentially risky scenario could happen when $S$ blocks the vision of $A$. In this scenario, $S$ is the parent of $A$, and $P$ and $A$ cause the collision $C$. $I$ does not influence other objects.

- **Highway**. An autonomous vehicle $A$ takes a lane-changing behavior due to a static car $S$ parked in front of it. Meanwhile, a vehicle $R$ drives in the opposite lane. Since $S$ blocks the vision of $A$, $A$ is likely to collide with $R$. In this scenario, $S$ is the parent of $A$, and $R$ and $A$ cause the collision $C$. $I$ does not influence other objects.

**Simulator.** We implement the above scenarios in a 2D simulator, where all agents have radar sensors and are controlled by a simple vehicle dynamic. During the running, the autonomous vehicle is controlled by a rule-based policy, which will decelerate if it detects any obstacles in front of it within a certain range Thus, the safety-critical scenario will not happen unless the radar of one agent is blocked and the distance is smaller than the braking distance, avoiding the creation of unrealistic scenarios. The action space contains the acceleration and steering of all objects, and the state space contains the position and heading of all objects and the status of traffic lights if applicable.

**Baselines.** We consider 7 algorithms as baselines, including 5 scenario generation methods and 2 variants of our *CausalAF*. Learning to collide (L2C) [5] uses a Bayesian network to describe the relationship between objects. Multi-modal Generation (MMG) [4] uses an adaptive sampler to increase sample diversity. STRIVE [16] learns traffic prior from datasets and uses adversarial optimization to generate risk scenarios. SAC is a standard RL algorithm using the objective as the reward function. To further investigate the contribution of COM and CVM, we design two variants that share the same network structure as *CausalAF*. Baseline does not use COM or CVM, and Baseline+COM only uses COM.

## 4.2 Results Discussion

**How does *CausalAF* perform on safety-critical scenario generation? (Q1)** We train all generation methods in 3 environments and report the final objective values in Table 1. We observe that *CausalAF* achieves the best performance among all methods. L2C performs better than MMG and SAC because it also considers the structure of the scenario. We also notice that both Baseline and Baseline+COM have performance drops compared to *CausalAF*, indicating that the COM and CVM modules contribute to the autoregressive generating process. Baseline+COM performs a little better

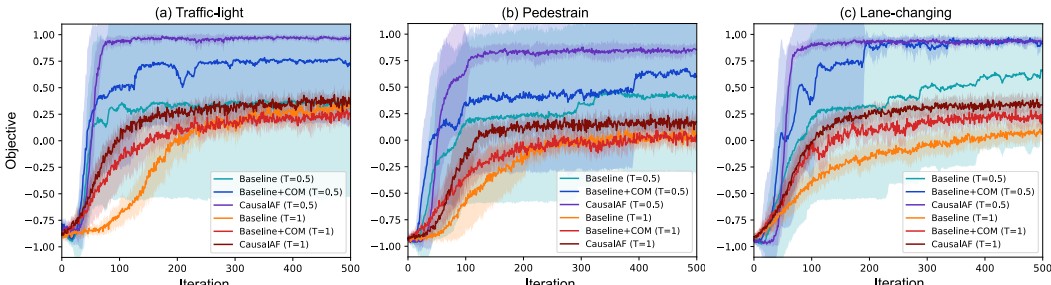

Figure 4: Training objective $\mathcal{L}_g(\mathcal{G}^B)$ of *CausalAF* and two variants under two sampling temperatures. The higher the sampling temperature is, the more diverse the generated scenarios are.

Table 2: Collision rate of RL algorithms evaluated in different scenarios.

| Method | Intersection | | | | Crossing | | | | Highway | | | |
|---|---|---|---|---|---|---|---|---|---|---|---|---|
| | Norm | L2C | MMG | Ours | Norm | L2C | MMG | Ours | Norm | L2C | MMG | Ours |
| SAC-Norm | 0.05 | 0.57 | 0.64 | 0.91 | 0.04 | 0.54 | 0.67 | 0.92 | 0.03 | 0.79 | 0.75 | 0.95 |
| SAC-Ours | 0.01 | 0.03 | 0.04 | 0.08 | 0.00 | 0.04 | 0.06 | 0.11 | 0.02 | 0.01 | 0.04 | 0.09 |
| PPO-Norm | 0.07 | 0.44 | 0.48 | 0.86 | 0.03 | 0.53 | 0.61 | 0.80 | 0.02 | 0.62 | 0.64 | 0.92 |
| PPO-Ours | 0.00 | 0.04 | 0.01 | 0.12 | 0.02 | 0.03 | 0.03 | 0.08 | 0.01 | 0.02 | 0.03 | 0.13 |
| DDPG-Norm | 0.12 | 0.76 | 0.62 | 0.89 | 0.07 | 0.71 | 0.76 | 0.85 | 0.04 | 0.72 | 0.61 | 0.95 |
| DDPG-Ours | 0.01 | 0.02 | 0.05 | 0.13 | 0.02 | 0.01 | 0.04 | 0.12 | 0.03 | 0.03 | 0.03 | 0.16 |
| MBRL-Norm | 0.04 | 0.78 | 0.74 | 0.98 | 0.05 | 0.68 | 0.85 | 0.97 | 0.05 | 0.79 | 0.87 | 0.98 |
| MBRL-Ours | 0.00 | 0.01 | 0.01 | 0.07 | 0.00 | 0.01 | 0.02 | 0.09 | 0.00 | 0.03 | 0.01 | 0.10 |

than Baseline, which validates our hypothesis that COM is not powerful enough to represent causality. To investigate the training procedure, we plot the training objectives in Figure 4 with two different sampling temperatures $T$, which controls the sampling variance in $\epsilon \sim \mathcal{N}(0, T)$. A large temperature provides strong exploration but causes slow convergence. However, we find that using a small temperature leads to unstable training with high variance due to poor exploration capability.

**How does causality help the generation process? (Q2)** The design of the Baseline represents the model that uses the full graph. Therefore, the results in Table 1 also demonstrate that the causal graph is more helpful than the full graph. To investigate the reason why the causal graph helps the learning, we conduct an ablation study on the number of irrelevant nodes ($I$ node), which do not have edges in the causal graph. In Figure 5, we can see that adding more irrelevant vehicles enlarges the gap between *CausalAF* and Baseline – the performance of Baseline gradually drops as the number of $I$ nodes increases but *CausalAF* has consistent performance. The reason is that *CausalAF* is able to diminish the impact of irrelevant information with COM and CVM.

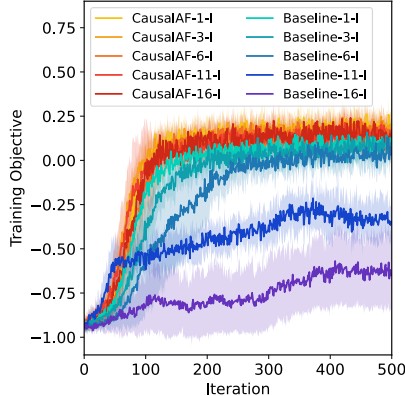

Figure 5: The training objectives in the Pedestrian scenario from different numbers of irrelevant vehicles.

**How can we use the generated scenarios? (Q3)** Finally, we explore how to use generated safety-critical scenarios. We train 4 RL agents ({SAC, PPO, DDPG, MBRL}-Norm) under normal scenarios (uniformly sample the parameters of objects in the scenario) then we evaluate them under scenarios generated by four different methods: Normal, L2C, MMG, and Ours (*CausalAF*) to test the performance under safety-critical scenarios. We also train another 4 agents under scenarios generated by our method ({SAC, PPO, DDPG, MBRL}-Ours) and evaluate under four different scenarios. We report the collision rate in Table 2. We find that scenarios generated by our CausalAF cause more collision to the RL agents, which also shows that training on normal scenarios is not enough for safety. After training on scenarios generated by CausalAF, the agents achieve lower collision in all scenarios, indicating the usefulness of training on safety-critical scenarios.

# 5  Related Work

**Goal-directed generative models.** DGMs, such as Generative Adversarial Networks [17] and Variational Auto-encoder [18], have shown powerful capability in randomly data generation tasks [19]. Among them, goal-directed generation methods are widely used [20]. One line of research leverages conditional GAN [21] and conditional VAE [22], which take as input the conditions or labels during the training stage. Another line of research injects the goal into the model after the training. [23] proposes a latent space optimization framework that finds the samples by searching in the latent space. This spirit is also adopted in other fields: [24] finds the molecules that satisfy specific chemical properties, [25] searches in the latent space of StyleGAN [26] to obtain targeted images. Recent works combine the advantages of the above two lines by iteratively updating the high-quality samples and retraining the model weights during the search [27]. [28] pre-trains the generative model and optimizes the sample distribution with reinforcement learning algorithms.

**Safety-critical driving scenario generation.** Traditional scenario generation algorithms sample from pre-defined rules and grammars, such as probabilistic scene graphs [29] and heuristic rules [30]. In contrast, DGMs [31, 32, 33, 34] are recently used to construct diverse scenarios. Adversarial optimization is considered for safety-critical scenario generation. [35, 36, 37] manipulate the pose of objects in traffic scenarios, [38, 39] adds objects on the top of existing vehicles to make them disappear, and [3] generates the layout of the traffic scenario with a tree structure integrated with human knowledge. Another direction generates risky scenarios while also considering the likelihood of occurring of the scenarios in the real world. [40, 41, 42] used various importance sampling approaches to generate risky but probable scenarios. [34] merges the naturalistic and collision datasets with conditional VAE. [43, 16, 44, 45] learn traffic prior from pre-collected dataset.

**Causal generative models.** The research of causality [8] is usually divided into two aspects: causal discovery finds the underlying mechanism from the data; causal inference extrapolates the given causality to solve new problems. A toolbox named NOTEARs is proposed in [46] to learn causal structure in a fully differentiable way, which drastically reduces the complexity caused by combinatorial optimization. [47] show the identifiability of learned causal structure from interventional data, which is obtained by manipulating the causal system under interventions. Recently, causality has been introduced into DGMs to learn the cause and effect with representation learning. CausalGAN [48] captures the causality by training the generator with the causal graph as a prior, which is very similar to our setting. In CausalVAE [49], the authors disentangle latent factors by learning a causal graph from data and corresponding labels. Previous work CAREFL [50] also explored the combination of causation and autoregressive flow-based model and is used for causal discovery and prediction tasks.

# 6  Conclusion and Limitation

This paper proposes a causal generative model that generates safety-critical scenarios with causal graphs obtained from humans prior. To incorporate the causality into the generation, we use the causal graph to regularize the generation of the behavioral graph, which is achieved by modifying the generating ordering and graph connection with two causal masks. By injecting causality into generation, we efficiently create safety-critical scenarios that are too rare to find in the real world. The experiment results on three environments with clear causality demonstrate that *CausalAF* outperforms all baselines in terms of efficiency and performance. We also show that training on our generated safety-critical scenarios improves the robustness of RL-based driving algorithms. The proposed method can be naturally extended to other robotics areas since critical scenarios are vital for learning-based algorithms but rare to collect in the real world, e.g., risky scenarios for household robots that involve human interaction.

The main limitation of this work is that the causal graph, summarized by humans, is assumed to be always correct, which may not be true for complicated scenarios. We will explore methods robust to human bias when attaining the causal graph, for example, automatically discovering causal graphs from observational or interventional datasets. Although this work is evaluated in simulations, we believe the autonomous driving area still benefits from safety-critical scenarios with abstracted representation, which shares a smaller sim-to-real gap compared to directly using raw sensor input.

**Acknowledgments**

We gratefully acknowledge support from the National Science Foundation under grant CAREER CNS-2047454 and support from the Manufacturing Futures Initiative at Carnegie Mellon University made possible by the Richard King Mellon Foundation.

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
