# OpenReview forum: "CausalAF: Causal Autoregressive Flow for Safety-Critical Driving Scenario Generation"
_robot-learning.org/CoRL/2022/Conference — CoRL 2022 Poster_

### Official Review · Reviewer_Gk4u · 2022-07-31

**Originality:** Good
**Technical Quality:** Poor
**Clarity Of Presentation:** Good
**Impact:** 2

**Recommendation:**

Weak Reject: I recommend rejecting the paper, but will not argue for my recommendation if the majority of other reviewers have a different opinion.

**Summary:**

The paper proposes CausalAF, a method to generate safety-critical scenarios by learning the behavior graph given the causal graph. The paper claims dramatic improvements on generating scenarios that meet the objective over data-driven baselines. Finally, they also show that the generated scenarios are beneficial for training RL policies in contrast to random scenario generation.

**Issues:**

See weaknesses section.

**Quality Of The Limitations Section:**

Limitations are not well addressed

**Reviewer Expertise:**

3: The reviewer is fairly confident that the evaluation is correct

**Robotics Focus:**

Relevant but unlikely to deploy to hardware in near future

**Strengths And Weaknesses:**

### Strengths
- The introduction captures very well the state of the self-driving industry. Scenario generation is a highly relevant task at the moment towards solving self-driving.
- The paper is well structured and the manuscript is fairly polished.
- Representing scenarios in a graphical way makes sense, and learning behavior graphs that satisfy the causal structure is a relevant problem.
- It is good to see that causality helps the generation process as covered in L238-251.

### Weaknesses
- Adversarial optimization is not treated as a first-class baseline. This is a constant throughout the introduction, the related work as well as the experiment. However, I believe that with the objective described in L160, methods such as STRIVE, KING and AdvSim (see citations below) would do extremely well. Unfortunately, such comparisons aren't available in the paper.
- In L82 it says that the edge matrix represents the sequential interaction between objects. I didn't understand what's the sequential part. As far as I understand, the only attrributes getting optimized are the initial position and velocity. This is what I understand from L89.
- It is not clear what is the exact parameterization/architecture of $\mathcal{F}_\phi$. Can it handle a variable number of actors in the scenario? Or a different module needs to be trained for every scenario?
- Expert knowledge and effort is required to provide the causal graph. While it is interesting to include prior knowledge in the problem, this route seems very time-consuming and that it will not generalize to more complex scenarios. While the paper acknowledges this limitation, I believe it hinders the applicability far too much for this work to be useful as it stands.
- The scenarios are extremely simple, and the baselines perhaps too weak. In terms of scenario generation, I'd be interested in seeing a comparison against adversarial optimization as well as simple heuristics based on time-to-collision in Table 1. Moreover, in Table 2 the impact of the generated scenarios in training RL policies is only compared to a random baseline, as opposed to comparing to the other baselines for scenario generation.
- Finally, in the limitations section it is naively stated that the sim-to-real gap is "almost negligible". The paper doesn't provide any support for this argument, and I believe is not true.

```
@inproceedings{rempe2022generating,
  title={Generating Useful Accident-Prone Driving Scenarios via a Learned Traffic Prior},
  author={Rempe, Davis and Philion, Jonah and Guibas, Leonidas J and Fidler, Sanja and Litany, Or},
  booktitle={Proceedings of the IEEE/CVF Conference on Computer Vision and Pattern Recognition},
  pages={17305--17315},
  year={2022}
}
@article{hanselmann2022king,
  title={KING: Generating Safety-Critical Driving Scenarios for Robust Imitation via Kinematics Gradients},
  author={Hanselmann, Niklas and Renz, Katrin and Chitta, Kashyap and Bhattacharyya, Apratim and Geiger, Andreas},
  journal={arXiv preprint arXiv:2204.13683},
  year={2022}
}
@inproceedings{wang2021advsim,
  title={Advsim: Generating safety-critical scenarios for self-driving vehicles},
  author={Wang, Jingkang and Pun, Ava and Tu, James and Manivasagam, Sivabalan and Sadat, Abbas and Casas, Sergio and Ren, Mengye and Urtasun, Raquel},
  booktitle={Proceedings of the IEEE/CVF Conference on Computer Vision and Pattern Recognition},
  pages={9909--9918},
  year={2021}
}
```

**Summary Of Recommendation:**

While I think the idea has merit, I believe removing the need for the expert-provided causal graphs and making more thorough experiments are a must to accept this paper. The baselines the method is compared against are weaker than they should.

---

> ### Author Response · Authors · 2022-08-24
> **Response to Reviewer Gk4u (1/3)**
>
> We thank the reviewer for valuable feedback and acknowledgment of the novelty of our method and the importance of our topic. We address the reviewer’s concerns in the following.
>
> ### **Q1: Adversarial optimization is not treated as a first-class baseline.**
>
> > However, I believe that with the objective described in L160, methods such as STRIVE, KING, and AdvSim (see citations below) would do extremely well. Unfortunately, such comparisons aren't available in the paper.
>
> Thanks for providing recent works as potential baselines. We want to mention that both L2C and MMG baselines are based on adversarial optimization. We add all recommended papers in the revised related work section.
> STRIVE, KING, and AdvSim all belong to adversarial generation and share a very similar idea, thus we select STRIVE as an additional baseline since it is recently open-sourced and seems more advanced.
>
> We use a rule-based planner to generate a synthetic dataset for training the encoder and decoder, then use adversarial optimization to generate safety-critical scenarios. Since we don't use the NuScenes dataset, we modify the code to adapt it to our simulator and generation process. Please refer to **Q5** for experiment results and discussion.
>
> ### **Q2: The edge matrix represents the sequential interaction between objects.**
>
> > I didn't understand what's the sequential part. As far as I understand, the only attributes getting optimized are the initial position and velocity. This is what I understand from L89.
>
> By sequential interaction, we mean the object starts from an initial position and follows a constant speed, which is stored in the edge attributes. Note that both of them are relative to the target node of the edge. Thus, it represents the sequential information of one object.
>
> ### **Q3: What is the exact parameterization/architecture of $\mathcal{F}_{\phi}$?**
>
> > Can it handle a variable number of actors in the scenario? Or a different module needs to be trained for every scenario?
>
> Thanks to the autoregressive generation, we are able to generate a variable number of nodes in the behavior graph. Thus, one of the biggest advantages of our method is that we only need to train one model to represent all scenarios with a variable number of actors.
>
> ### **Q4: Expert knowledge and effort is required to provide the causal graph.**
>
> > While it is interesting to include prior knowledge in the problem, this route seems very time-consuming, and it will not generalize to more complex scenarios. While the paper acknowledges this limitation, I believe it hinders the applicability far too much for this work to be useful as it stands.
>
> We agree that the requirement of the causal graph may introduce extra human effort. However, using human knowledge to improve learning-based methods is absolutely an important direction. Particularly, previous methods in the safety-critical scenario generation area focus on data-driven modeling or adversarial attack. Due to the rareness of risk scenarios, both of them have large limitations, e.g., low diversity and inefficiency. Human knowledge, represented by the causal graph in this paper, is easily obtained from experienced drivers and provides valuable guidance to the generation process. We believe incorporating human knowledge has large potential to fundamentally solve the safety-critical scenario generation problem.
>
> In addition, causal graphs can also be obtained from other resources. There are already numerous works of causal discovery, which automatically find the true causal graph from observational and interventional data [1][2][3][4]. Jointly optimizing the generative model and the discovery model could be one promising extension of this work.

---

> > ### Author Response · Authors · 2022-08-24
> > **Response to Reviewer Gk4u (2/3)**
> >
> > ### **Q5: The scenarios are extremely simple, and the baselines perhaps too weak.**
> >
> > > In terms of scenario generation, I'd be interested in seeing a comparison against adversarial optimization as well as simple heuristics based on time-to-collision in Table 1. Moreover, in Table 2 the impact of the generated scenarios in training RL policies is only compared to a random baseline.
> >
> > We respectfully argue that the three template scenarios considered in this paper are non-trivial and realistic in the real world. One important point is that the complexity of traffic scenarios does not indicate the risk level. Most safety-critical scenarios happen on simple road shapes with only two or three objects. Although some previous works try to find risky scenarios from real-world data and road maps, we believe the important factors behind risky scenarios are logic and causality rather than squeezing the drivable area of the ego vehicle.
> >
> > To provide a more thorough comparison, we add two additional experiments.
> > 1. In Table 1, we add STRIVE for safety-critical scenario generation.
> >
> > **Table 1: Collision rate of generated safety-critical scenarios**
> > | Method| L2C|MMG|SAC|STRIVE| Baseline  | Baseline+COM | CausalAF |
> > | :---| :---: | :---: | :---: | :---: | :---: | :---: | :---: |
> > | Intersection| 0.63±0.28| 0.31±0.54 | 0.47±0.61| 0.64±0.12 | 0.29±0.84 | 0.69±0.52| 0.98±0.01 |
> > | Crossing| 0.69±0.41| 0.43±0.56 | 0.38±0.49| 0.55±0.10 | 0.35±0.65 | 0.57±0.48| 0.83±0.13 |
> > | Highway| 0.85±0.10| 0.56±0.36 | 0.58±0.41| 0.67±0.16 | 0.53±0.69 | 0.88±0.04| 0.91±0.06 |
> >
> > Three reasons to explain the gap between STRIVE and CausalAF:
> > * given the causal graph, the generative model ignores the nodes that are not relevant to the safety-critical events, e.g., vehicles in the opposite direction.
> > * CausalAF learns the causality behind the collision rather than memorizing the relative position between vehicles.
> > * STRIVE optimizes in a latent space, which is usually trained with normal data. So most risky scenarios generated by the adversarial optimization are obtained by manipulating the trajectories to have spatial-temporal overlap. However, in our three template scenarios, there exists causality behind the collision, e.g., the view of ego is blocked by another vehicle and thus fails to decelerate.
> >
> >
> > 2. In Table 2, we add additional experiments to evaluate the generated scenarios. We consider two different settings, XXX-Norm: train 4 RL agents in normal scenarios; XXX-Ours: train 4 RL agents in scenarios generated by our CausalAF. Then we test them in scenarios generated by Norm, L2C, MMG, and our CausalAF. We summarize the results below and also update the table in the revised manuscript.
> >
> > **Table 2: Collision rate of RL algorithms evaluated in different scenarios**
> > | Environment | Intersection  | | | | Crossing || | | Highway| | ||
> > | :--- | :---: | :--: | :--: | :---: | :---: | :--: | :--: | :---: | :---: | :--: | :---: | :---: |
> > | **Method**| **Norm** | **L2C** | **MMG** | **Ours** | **Norm** | **L2C** | **MMG** | **Ours** | **Norm** | **L2C** | **MMG** | **Ours** |
> > | SAC-Norm | 0.05     | 0.57  | 0.64    | 0.91     | 0.04     | 0.54    | 0.67    | 0.92     | 0.03     | 0.79    | 0.75    | 0.95     |
> > | SAC-Ours | 0.01     | 0.03    | 0.04    | 0.08     | 0.00     | 0.04    | 0.06    | 0.11     | 0.02     | 0.01    | 0.04    | 0.09     |
> > | PPO-Norm | 0.07     | 0.44    | 0.48    | 0.86     | 0.03     | 0.53    | 0.61    | 0.80     | 0.02     | 0.62    | 0.64    | 0.92     |
> > | PPO-Ours | 0.00     | 0.04    | 0.01    | 0.12     | 0.02     | 0.03    | 0.03    | 0.08     | 0.01     | 0.02    | 0.03    | 0.13     |
> > | DDPG-Norm| 0.12     | 0.76    | 0.62    | 0.89     | 0.07     | 0.71    | 0.76    | 0.85     | 0.04     | 0.72    | 0.61    | 0.95     |
> > | DDPG-Ours | 0.01     | 0.02    | 0.05    | 0.13     | 0.02     | 0.01    | 0.04    | 0.12     | 0.03     | 0.03    | 0.03    | 0.16     |
> > | MBRL-Norm | 0.04     | 0.78    | 0.74    | 0.98     | 0.05     | 0.68    | 0.85    | 0.97     | 0.05     | 0.79    | 0.87    | 0.98     |
> > | MBRL-Ours | 0.00     | 0.01    | 0.01    | 0.07     | 0.00     | 0.01    | 0.02    | 0.09     | 0.00     | 0.03    | 0.01    | 0.10     |
> >
> > We find that scenarios generated by CausalAF cause more collision to XXX-Norm agents, which indicates that risky scenario generation methods indeed cause more collision compared to normal scenarios. Note that our method causes more collision to RL agents. The main reason is that all scenario generators are trained against a rule-based planner. By using the causal graph, CausalAF learns the causality behind the collision rather than memorizing the relative position between vehicles as L2C and MMG do.
> >
> > After training on scenarios generated by CausalAF, XXX-Ours agents dramatically reduce the collision rates under safety-critical scenarios generated by all methods. This means that training in safety-critical scenarios can improve the safety of RL algorithms.

---

> > > ### Author Response · Authors · 2022-08-24
> > > **Response to Reviewer Gk4u (3/3)**
> > >
> > > ### **Q6: It is not true that the sim-to-real gap is "almost negligible".**
> > >
> > > > Finally, in the limitations section it is naively stated that the sim-to-real gap is "almost negligible". The paper doesn't provide any support for this argument, and I believe is not true.
> > >
> > > We agree that the sim-to-real gap is still not negligible. To avoid overstatement, we rewrite the sentence as "we believe the autonomous driving area still benefits from safety-critical scenarios with abstracted representation, which shares a smaller sim-to-real gap compared to directly using raw sensor input."
> > >
> > > We also want to elaborate a little bit more for consideration in our original arguments.
> > > Since modern simulators usually have accurate models of dynamics, the decision-making algorithms are usually easy to be transferred from simulation to the real world. In contrast, the quality of perception in simulators is still very different from the real world, which usually results in large sim-to-real gaps.
> > > As our paper is built upon a scenario description with low-level information (e.g. vehicle trajectories and velocities), we can avoid the large sim-to-real gaps from the perception.
> > >
> > > ---
> > >
> > > [1] Vowels, M.J., Camgoz, N.C. and Bowden, R., 2021. D’ya like DAGs? A survey on structure learning and causal discovery. ACM Computing Surveys (CSUR).
> > >
> > > [2] Yu, Y., Chen, J., Gao, T. and Yu, M., 2019, May. DAG-GNN: DAG structure learning with graph neural networks. In International Conference on Machine Learning (pp. 7154-7163). PMLR.
> > >
> > > [3] Brouillard, P., Lachapelle, S., Lacoste, A., Lacoste-Julien, S. and Drouin, A., 2020. Differentiable causal discovery from interventional data. Advances in Neural Information Processing Systems, 33, pp.21865-21877.
> > >
> > > [4] Lippe, P., Cohen, T. and Gavves, E., 2021. Efficient neural causal discovery without acyclicity constraints. arXiv preprint arXiv:2107.10483.

---

> ### Author Response · Authors · 2022-08-28
> **Any further questions?**
>
> Dear reviewer,
>
> We provide detailed responses to your questions and additional experiment results to show the advantages of our method. We kindly ask the reviewer to reassess the paper in light of our response. If all concerns are addressed, please consider re-evaluating the recommendation score. If not, we are very glad to have further discussions. Thanks!

---

### Official Review · Reviewer_pK7P · 2022-07-31

**Originality:** Very Good
**Technical Quality:** Very Good
**Clarity Of Presentation:** Good
**Impact:** 3

**Recommendation:**

Weak Accept: I recommend accepting the paper, but will not argue for my recommendation if the majority of other reviewers have a different opinion.

**Summary:**

The objective of this paper is to generate test cases for automated driving systems that present realistic and difficult situations (which are only rarely occurring under generic sampling methods).  The idea is to augment a generative model for sequences of important events with constraints from a causal model, thus generating more plausible scenarios that still result in a high probability of impact and that therefore represent good test problems.

**Issues:**

I wouldn't ask for any additional work during the revision period.  I think the rewriting required is fairly substantial.

**Quality Of The Limitations Section:**

Limitations are addressed clearly

**Reviewer Expertise:**

2: The reviewer is willing to defend the evaluation, but it is quite likely that the reviewer did not understand central parts of the paper

**Robotics Focus:**

Highly relevant to robotics but no hardware experiments

**Strengths And Weaknesses:**

The problem addressed by the paper is important and the ideas seem sensible.  In particular, this method provides us with a way of taking some human expertise (basic, generic causal structures) in a form that is easy for humans to supply and leveraging it to generate good test scenarios.

One possible objection is that this work is not too relevant to the field of robotics;  but it is relevant to the extent that automated vehicles are robots and that it is critical to train them on data distributions that exhibit important possible failure modes.

An answer to this objection which (I think) would actually make the paper stronger, would be to assert that this work is important to any agent that is being trained via typical learning methods to take actions in a complex domain in which there are seriously risky situations that are relatively rare, and that these situations can come up in a wide variety of robotics applications (including industrial or household robots sharing workspace with humans, or robots working on disaster recovery.)

A primary difficulty with this paper is clarity and organization of the writing.  It is very important to describe, rigorously and early,   exactly what kind of testing data you will produce and how it is used.
- What exactly is the type of a scenario $\tau$?  Does it specify detailed trajectories for all the vehicles, stop-lights, etc?   Or all but the agent?
- How are the scenarios used during reinforcement learning?   Do we use the current RL policy $\pi$ inside $\mathcal E$ to generate new scenarios, and then do offline RL in those scenarios to generate a new policy?
- Are the constraints from the causal graph imposed *at scenario generation time* or are they somehow incorporated into $\phi$ so that the underlying autoregressive generation process tends to generate graphs that satisfy them?

Here are some other questions that came up for me as I was reading, which might be addressable by rewriting, taking care to make sure the paper is readable by someone new to your work, defining terms before using them, etc.
- What does it mean for a graph to "work as a high-level planner"?
- How does the behavioral graph encode the ordering of actions given that there can be parallel self-loops?   How do you guarantee the actions are legal?
- It would be better to explain the behavioral graph (semantics and use) before talking about how to generate them.
- Also, explain to us how the behavioral graph will help generate rare-but-interesting scenarios.  (I finally understood this later in the paper)
- 2.2 Say earlier that G^B is a DAG.
- Are CG and BG the same type?  What does it mean to minimize distance between them?
- How can a node be directed?
- The description of causal visible masks could be clearer.  It "filters the non-causal information when generating edges"  But what does it
mean to do that?
- Doesn't the likelihood of collision depend on the policy being executed?  I guess you are going to do this dynamically?  Given a
policy, generate scenarios that cause crashes, use those scenarios to improve your policy, etc.?
- It is generally good practice to only put into bold the method(s) that are statistically significantly best or indistinguishable from best (not just with the highest mean)
- Are these scenarios necessarily interesting if they cause a simple rule-based policy to crash?  But I guess the idea is that this method can be used
to cause other policies to crash, as well?
- I like that you put in a pedagogical figure (2) but for it to actually teach, it would need more explanation.

The study on irrelevant nodes was good and illuminating.

It would be great to explain more fully the importance of having monotonicity of likelihood.

This whole framework of auto-regressive graph generation is very cool, and adding causal constraints is a nice idea.   Might there be other kinds of constraints that humans could specify (or "skeleton" scenarios) that could also be incorporated in this framework?

-------------------
After discussion, I feel that:
- the exposition is improved but could probably be made more systematic still
- the solution really is interesting and feels novel
- I still don't quite understand what you mean by:
"Nodes in behavior graph represent objects we are interested in, e.g., vehicles, pedestrians, traffic signs, traffic lights, etc. Edges in represent the properties of objects, e.g., initial positions, velocities, and colors. Therefore, the behavior graph describes the sequential interaction between objects, which can be viewed as a motion planner."
To me, a "planner" takes some kind of goal or objective and some kind of transition model and does some search to find an action sequence that optimizes the objective.

I will improve my recommendation


**Summary Of Recommendation:**

The ideas here seem interesting but are not really clearly enough articulated for me to have understood the proposed methods well enough for me to evaluate them.  I would encourage a rewrite of the paper.

---

> ### Author Response · Authors · 2022-08-24
> **Response to Reviewer pK7P (1/2)**
>
> We thank the reviewer for finding our topic important and our ideas sensible. We also thank the reviewer for providing so many suggestions to improve the clarification of our paper. We carefully address all the questions and revise the manuscript according to the answers.
>
> ### **Q1: This work is not too relevant to the field of robotics.**
>
> > One possible objection is that this work is not too relevant to the field of robotics, but it is relevant to the extent that automated vehicles are robots and that it is critical to train them on data distributions that exhibit important possible failure modes.
> An answer to this objection which (I think) would actually make the paper stronger, would be to assert that this work is important to any agent that is being trained via typical learning methods to take actions in a complex domain in which there are seriously risky situations that are relatively rare and that these situations can come up in a wide variety of robotics applications (including industrial or household robots sharing workspace with humans, or robots working on disaster recovery.)
>
> We totally agree that generating safety-critical scenarios is important and useful in lots of robotics areas besides autonomous driving. For example, our method can be easily used for household mobile robots and robot arms. We add an additional discussion about the natural extension of the proposed method to other robotics applications in the Conclusion section. We believe autonomous driving could be a good start since it is one kind of mobile robot that has a big chance to be deployed in the real world.
>
> ### **Q2: What kind of testing data you will produce and how it is used?**
>
> > What exactly is the type of scenario $\tau$ ? Does it specify detailed trajectories for all the vehicles, stop-lights, etc? Or all but the agent?
>
> In one word, a scenario $\tau$ contains all information that happens when running the simulator with a behavior graph $\mathcal{G}^B$, including the trajectories of actors, the color of traffic lights, etc.
> A behavior graph $\mathcal{G}^B$ describes the properties of objects in the scenario. Nodes in $\mathcal{G}^B$ represent objects we are interested in, e.g., vehicles, pedestrians, traffic signs, traffic lights, etc. Edges in $\mathcal{G}^B$ represent the properties of objects, e.g., initial positions, velocities, and colors. Different types of objects have different properties.
> Then, a scenario $\tau$ is generated by executing $\mathcal{G}^B$ in the simulator, i.e., running the vehicles and changing the color of traffic light according to the properties defined in $\mathcal{G}^B$.
>
> > How are the scenarios used during reinforcement learning? Do we use the current RL policy $\pi$ inside $\mathcal{E}$ to generate new scenarios, and then do offline RL in those scenarios to generate a new policy?
>
> Generated scenarios are used as environments for RL methods. In the traditional RL setting, the scenarios are usually randomly generated with surrounding vehicles following pre-defined trajectories. In our case, we first train the CausalAF model with rule-based ego vehicles to learn how to generate risky scenarios and fix the parameters. Then, we run the RL policy $\pi$ together with a behavior graph $\mathcal{G}^B$ (generated by CausalAF) in the simulator. Therefore, the RL agent always runs in safety-critical scenarios instead of random scenarios.
>
> > Are the constraints from the causal graph imposed at scenario generation time or are they somehow incorporated into $\phi$ so that the underlying autoregressive generation process tends to generate graphs that satisfy them?
>
> The constraints are used during the training time. Since we use two types of causal masks during the training to encourage the generated results to satisfy the constraints, the learned parameters $\phi$ implicitly contain the information of cause and effect. In the generation stage, we still use the constraints to make sure the model has the same causal structure as the training stage.

---

> > ### Author Response · Authors · 2022-08-24
> > **Response to Reviewer pK7P (2/2)**
> >
> > ### **Q3: Some other questions that might be addressable by rewriting.**
> >
> > > What does it mean for a graph to "work as a high-level planner"?
> >
> > Nodes in behavior graph $\mathcal{G}^B$ represent objects we are interested in, e.g., vehicles, pedestrians, traffic signs, traffic lights, etc. Edges in $\mathcal{G}^B$ represent the properties of objects, e.g., initial positions, velocities, and colors. Therefore, the behavior graph describes the sequential interaction between objects, which can be viewed as a motion planner.
> >
> > > How does the behavioral graph encode the ordering of actions given that there can be parallel self-loops? How do you guarantee the actions are legal?
> >
> > In the generated behavioral graph, one node can only have one self-loop. So there will exist no parallel self-loop and all actions are executed at the same time. To make all actions legal and executable, we design some masks during the generation. If one action is not legal for one node, the probability of this action in the categorical distribution will be set to 0.
> >
> > > It would be better to explain the behavioral graph (semantics and use) before talking about how to generate them.
> >
> > Thank you for providing this suggestion, we add a simple usage of the behavioral graph in Section 2.1 and add the detailed definition of nodes and edges used in our experiment in Appendix.B.2.
> >
> > > 2.2 Say earlier that G^B is a DAG.
> >
> > We update the definition of $\mathcal{G^B}$ in Section 2.1.
> >
> > > Are CG and BG the same type? What does it mean to minimize distance between them?
> >
> > Both CG and BG are directed graphs, so the distance here means the similarity of the structure instead of the attribute. We change the word "distance" to "structure similarity".
> >
> > > How can a node be directed?
> >
> > Sorry for the typo, what we mean is "directed edges". We've changed this in the modified version.
> >
> > > The description of causal visible masks could be clearer. It "filters the non-causal information when generating edges" But what does it mean to do that?
> >
> > We rewrite this sentence in the manuscript: "We further propose another type of mask called CVM, which removes the non-causal connections, i.e., non-parent nodes to the current node in $\mathcal{G}^C$, when generating edges."
> >
> > > Doesn't the likelihood of collision depend on the policy being executed? I guess you are going to do this dynamically? Given a policy, generate scenarios that cause crashes, use those scenarios to improve your policy, etc.?
> >
> > Yes, the collision is calculated based on executed policy. In our experiment, we use a rule-based policy (uses PID controller for trajectory following and brakes when encounters obstacles) as a surrogate model to obtain the likelihood. Because of the generalizability. the generated scenarios are still safety-critical for other different policies. The iterative training of generator and policy is usually used for developing a better policy model, which is not considered in this paper. However, this is definitely one usage of our method.
> >
> > > Are these scenarios necessarily interesting if they cause a simple rule-based policy to crash? But I guess the idea is that this method can be used to cause other policies to crash, as well.
> >
> > Yes, we find the safety-critical scenarios have good generalizability, which means they can also cause collisions with other policies that are not used during the training. This finding is supported by the RL experiment, where the scenarios target rule-based policies but still fail RL methods.
> >
> > > I like that you put in a pedagogical figure (2) but for it to actually teach, it would need more explanation.
> >
> > We add more explanation in the caption of Figure 2.
> >
> > ### **Q4: It would be great to explain more fully the importance of having monotonicity of likelihood.**
> >
> > Firstly, the guarantee of monotonicity of likelihood ensures CausalAF's optimality under the true causal graph, i.e. whenever we have the true causal graph, CausalAF converges at the maximum likelihood.
> > Secondly, this can also be extended to the circumstances where we don't have a ground truth causal graph. By conducting automatic causal discovery, if the new causal graph shares more structure similarity to the ground truth, the performance of CausalAF should also improve. As an inspiring future direction, under the monotonicity guarantee, we plan to further introduce a causal discovery module to the current safety-critical scenario generation frameworks.
> >
> > ### **Q5: Might there be other kinds of constraints that humans could specify (or "skeleton" scenarios) that could also be incorporated into this framework?.**
> >
> > Thanks for asking this inspiring question. Actually, the causal graph also serves as a template scenario. When humans design the causal graph, they need to specify what kind of object (node type) they are interested in and what kind of road map they consider. So the causal graph already contains some important information about the "skeleton" of the scenario.

---

> ### Author Response · Authors · 2022-08-28
> **Any further questions?**
>
> Dear reviewer,
>
> We provide detailed responses to your questions and additional experiment results to show the advantages of our method. We kindly ask the reviewer to reassess the paper in light of our response. If all concerns are addressed, please consider re-evaluating the recommendation score. If not, we are very glad to have further discussions. Thanks!

---

### Official Review · Reviewer_9o2A · 2022-07-31

**Originality:** Very Good
**Technical Quality:** Very Good
**Clarity Of Presentation:** Very Good
**Impact:** 4

**Recommendation:**

Strong Accept: I recommend accepting the paper and will argue for my recommendation even if other reviewers hold a different opinion.

**Summary:**

This paper presents a solution to generating scenarios in an efficient manner (targeted to automated driving, but more generally applicable).  The authors present a means for leveraging causality, in particular causal graphs, in constructing a behavior graph that can serve to instantiate scenarios.  The sample efficiency comes from the behavior graph, which alleviates the need for all combinations of behaviors, even implausible ones, that may occur with standard generative models.

**Issues:**

1. Addressing what redeeming qualities are for autoregressive flows over other types of normalizing flows, any background on why normalizing flows and its impact on likelihood computation, as this is not explained anywhere.
2. Regarding causal graphs, what are the limitations in terms of diversity of the scenarios if the user is expected to provide such domain-specifics?  The authors should provide an empirical justification for causal graphs, and improve the discussion on this topic.
3. The authors should address whether there are inherent limitations in the causal graphs and the generated behavior graphs.  For instance, are bi-directional interactions (one agent influences another, and vice versa) possible or not?
4. A better background and discussion on behavior graphs and causal graphs.  For instance, better descriptions of what "type" means in context, by way of better examples.
5. How are $V_i$, $E_{i,j}$ initialized in the algorithm 1?
6. There are a number of grammar mistakes. For instance, inconsistent references to figures ("Figure X" vs. "Fig X"), and sentence fragments ("... merges the naturalistic and collision datasets with conditional." on line 282).

**Quality Of The Limitations Section:**

Additional details required

**Reviewer Expertise:**

4: The reviewer is confident but not absolutely certain that the evaluation is correct

**Robotics Focus:**

Highly relevant to robotics but no hardware experiments

**Strengths And Weaknesses:**

The paper is well-written, offering a unique take on an important problem of finding rare events and corner cases for AV safety in a principled way.  The presented approach seems to offer a definitive improvement over existing methods in terms of sample efficiency, in cases where causality can be leveraged.  The authors introduce a novel procedure for using AF to construct a behavior graph in generating two types of causal masks: a causal order mask, and a causal visible mask to properly impose causal ordering of nodes and edges.  The authors also include a well-written code artifact, which will certainly boost reproducibility.

One major weakness of the paper is the rather shallow descriptions motivating the various components, particularly in generating the causal graph and autoregressive flows.  It seems quite a bit counterintuitive that to generate a simulation of a rare event or rare scenario, that the details of such a scenario must already be specified by a human (i.e. the causal graph must be known).  Indeed, in all such scenarios It would be worth elaborating on why such a scheme helps, starting with some example scenarios.

There is also little background given on the choice of autoregressive flows, and no introduction to normalizing flows.  What are the redeeming properties of autoregressive flows?

Some further questions on the implications of using causal graphs.  For both the causal graph and behavioral graph, do they allow for two-way interactions between agents?  For instance, the case of two vehicles negotiating a merge?  From the definitions given, this does not seem to be the case.  More generally, the authors are strongly encouraged to note any inherent limitations in the causal graph that may make it impossible to form scenarios that may otherwise be possible using causality-free approaches.  Also, the node and edge types are also not clear - what do the authors mean by “type”?  Further explanations in the behavioral graph section (Section 2.1) should be added, and examples should be more complete.

While I acknowledge that the authors do address the causal graph requirement as a limitation, I would recommend that authors strongly consider adding an empirical exploration on the diversity of the generated scenarios on a limited set of causal graphs versus the diversity of causality-free methods.  In other words, how much diversity is lost by relying on users to supply a causal graph?

**Summary Of Recommendation:**

Because this paper presents a novel approach to scenario generation, with clear empirical benefits, I would advocate for acceptance.  The algorithmic contributions make for a very strong contribution, as does the code that is released along with the paper. The authors should address the issues brought forth in this review; in particular, by better motivating the use of autoregressive flows and explaining further the limitations of having a causal graph, particularly in loss of scenario coverage (diversity).  Further empirical studies on the diversity issue are highly encouraged.

---

> ### Author Response · Authors · 2022-08-24
> **Response to Reviewer 9o2A**
>
> We thank the reviewer for valuable feedback and acknowledgment of the novelty of our method. We address the reviewer’s concerns in the following.
>
> ### **Q1: Shallow descriptions motivating the autoregressive flow**
>
> > Addressing what redeeming qualities are for autoregressive flows over other types of normalizing flows, any background on why normalizing flows and its impact on likelihood computation, as this is not explained anywhere.
>
> We add the motivation of using autoregressive normalizing flow in Section 2.2 of the revised manuscript. We also summarize the main reasons here:
>
> * The autoregressive flow has an iterative sampling process, which allows injecting causal knowledge by using COM and CVM.
>
> * The traffic scenarios usually contain a varying number of vehicles. During training, autoregressive flow can be trained in parallel on scenarios with different sequence lengths. During generation, autoregressive flow also can generate scenarios with different numbers of vehicles.
>
> ### **Q2: how much diversity is lost by relying on users to supply a causal graph**
>
> > Regarding causal graphs, what are the limitations in terms of the diversity of the scenarios if the user is expected to provide such domain-specifics? The authors should provide an empirical justification for causal graphs, and improve the discussion on this topic.
>
> Thanks for asking this inspiring question. Generally, there exists a trade-off between diversity and likelihood in generative models. In our model, we inject a strong inductive bias, i.e., the causal graph, into the generation process, which may restrict the scenarios to a subset of all scenarios. However, our model still has a large room for diversity since the causal graph only restricts the general interaction between vehicles and pedestrians rather than all scenario parameters. On the other hand, the users only need to give a rough causal graph to form a "template" scenario, our CausalAF model will generate diverse safety-critical scenarios based on it.
>
> We provide additional analysis of the distribution of scenario parameters by comparing two models, one with the causal graph and the other without. The results are shown in Appendix.D.2.
>
> ### **Q3: Inherent limitations in the causal graph design.**
>
> > The authors should address whether there are inherent limitations in the causal graphs and the generated behavior graphs. For instance, are bi-directional interactions (one agent influences another, and vice versa) possible or not?
>
> Thanks for asking this clarification question. We agree that there are some limitations to the behavior graph. Since we assume the safety-critical scenarios happen because of the influence of one object on another object, our behavior graph only considers one direction to match the definition of the causal graph.
>
> ### **Q4: the node and edge types are also not clear**
>
> > A better background and discussion on behavior graphs and causal graphs. For instance, better descriptions of what "type" means in context, by way of better examples.
>
> We provide more details about the causal graphs and behavior graphs in Appendix B.2, including the types and attributes of nodes and edges.
>
> ### **Q5: How are $V_i$ and $E_{i,j}$ initialized in the algorithm 1?**
>
> The initial behavior graph is empty, therefore the node vector $V$ and adjacency matrix $E$ are initialized with zeros.
>
> ### **Q6: There are a number of grammar mistakes**
>
> Thanks for carefully checking the grammar of our paper. We revised the manuscript with updates on grammar and inconsistent references.

---

> ### Author Response · Authors · 2022-08-28
> **Any further questions?**
>
> Dear reviewer,
>
> We provide detailed responses to your questions and additional experiment results to show the advantages of our method. We are glad to have a discussion if you have further questions. Thanks!

---

### Official Review · Reviewer_aF4L · 2022-08-05

**Originality:** Very Good
**Technical Quality:** Very Good
**Clarity Of Presentation:** Good
**Impact:** 4

**Recommendation:**

Weak Accept: I recommend accepting the paper, but will not argue for my recommendation if the majority of other reviewers have a different opinion.

**Summary:**

Although nowadays autonomous vehicles can handle most scenarios, their performance in a small number of safety-critical cases is not evaluated comprehensively. The main reason is that these cases are difficult to collect and thus the number is limited. This paper proposes a new generative method taking causal graphs as input to produce countless safety-critical scenarios, which could help tame the long tail problem and help improve the safety of ADS. Extensive experiments are conducted to verify the effectiveness the proposed method.

**Issues:**

See weakness

**Quality Of The Limitations Section:**

Additional details required

**Reviewer Expertise:**

3: The reviewer is fairly confident that the evaluation is correct

**Robotics Focus:**

Relevant but unlikely to deploy to hardware in near future

**Strengths And Weaknesses:**

**Strength**:
1) This paper proposes a new graph-based scenario description tool so that causality could be integrated
2) The idea of generating scenarios from a causality graph is original and interesting
3) It seems that the authors give a solid convergence guarantee, based on the delicate notation. But I am not an expert at the graph model, and thus can not ensure the correctness of the modeling.
4) The experiment is complete, including baseline comparisons, ablation study (on COM and CVM) and RL experiments.

**Weakness**:
The weakest part of this paper is that the experiment setting should be made more clear and some qualitative results should be added.

1) What do the scores in Table 1 mean? Is it the collision rate of the generated case?
2) What do the generated scenarios look like? Some qualitative results should be visualized to show the diversity of generated scenarios.
3) Are there only 3 synthetic maps? If not, please describe how many maps are contained in these simulators and how do they look like, since the map structure will greatly influence the generated scenarios. If these maps are synthetic, I would like to know if there are any future plans to extend this method to real maps, which are more complex and unstructured
4) The pre-defined types of nodes and edges used in the experiment should be made clear
5) In the RL experiments, are the safety-critical cases split into training and test sets?





**Summary Of Recommendation:**

I think the weakness mentioned above is easy to solve in revision. Therefore, I would like to give a weak acceptance.

---

> ### Author Response · Authors · 2022-08-24
> **Response to Reviewer aF4L**
>
> We thank the reviewer for valuable feedback and acknowledgment of our experiment design. We address the reviewer’s concerns in the following.
>
> ### **Q1: The experiment setting should be made more clear and some qualitative results should be added.**
>
> We provide answers to the following questions in the revised manuscript in blue color.
>
> > What do the scores in Table 1 mean? Is it the collision rate of the generated case?
>
> The score represents the collision rate of the generated scenarios. We give the agent 1 reward when it causes collisions and 0 reward otherwise.
>
> > What do the generated scenarios look like? Some qualitative results should be visualized to show the diversity of generated scenarios.
>
> We provide three generated safety-critical scenarios in Appendix.D.1, where each row represents one environment. To show the diversity of generated scenarios, we plot the variances of position and velocity of vehicles and pedestrians in Appendix.D.2.
>
> > Are there only 3 synthetic maps? If not, please describe how many maps are contained in these simulators and how do they look like, since the map structure will greatly influence the generated scenarios. If these maps are synthetic, I would like to know if there are any future plans to extend this method to real maps, which are more complex and unstructured.
>
> We consider 3 representative maps in our experiments, including one highway, one intersection, and one crossing. Our method can be easily extended to real-world maps.
> One important point is that the complexity of traffic scenarios does not indicate the risk level. Most safety-critical scenarios happen in simple road maps with only two or three vehicles.
> Since the safety-critical scenarios usually occur because of the dynamic interaction between objects rather than the static road shape, we believe the efficiency and effectiveness of our method won't have many differences.
>
> > The pre-defined types of nodes and edges used in the experiment should be made clear
>
> We add a Table in Appendix.B.2 to introduce the definitions of nodes and edges in detail.
>
> > In the RL experiments, are the safety-critical cases split into training and test sets?
>
> No, we train and test the agents on the same safety-critical scenarios, which is the common setting in RL. However, testing the generalization of RL agents in different safety-critical scenarios is an interesting and important topic. We plan to explore this direction in the next step.

---

> > ### Comment · Reviewer_aF4L · 2022-08-25
> > **Thanks**
> >
> > Thank you for answering my questions. The revised version looks complete and I would like to change my recommendation to **accept**.

---

> > > ### Author Response · Authors · 2022-08-28
> > > **Thanks for rasing the score**
> > >
> > > We are glad our response and revision addressed your concerns. Thanks for pricing valuable suggestions to improve our work!

---

### Author Response · Authors · 2022-08-24
**The highlight of the revised manuscript**

We thank all the reviewers for their time and valuable suggestions. Following the suggestions, we have conducted additional experiments, clarified important settings, and rearranged key statements in our revision. We highlight our major updates below.

* We add STRIVE [1] as a new baseline for comparison in Table 1.

* We add an adversarial training experiment to show the usage of generated scenarios in Table 2.

* We provide qualitative examples of scenarios in Figure 6 and the analysis of the diversity of scenarios in Figure 7.

* We add a future plan of how to reduce human effort by the causal discovery in Section 6.

* We add a discussion about extending this work to other robotics fields in Section 6.

We also made some important changes based on specific questions from reviewers, you may check them in the revised manuscripts and our responses. We welcome all of you to join the discussion in the remaining days of the rebuttal phase!

---

[1] Rempe, D., Philion, J., Guibas, L.J., Fidler, S. and Litany, O., 2022. Generating Useful Accident-Prone Driving Scenarios via a Learned Traffic Prior. In Proceedings of the IEEE/CVF Conference on Computer Vision and Pattern Recognition (pp. 17305-17315).

---

### Meta-Review · Area_Chair_4qe5 · 2022-08-03

**Recommendation:** Accept (Poster)
**Confidence:** 4

**Metareview:**

Below is a summary of the strengths and weaknesses. Please see the reviews for more information, as they provide detailed feedback both on these points and on other points.

Strengths:
- The problem setting is interesting and relevant to the CoRL community (though the paper could also convey how these ideas may be relevant beyond autonomous driving to any robot that may encounter risky situations that are rare)
- The method is sensible and provides a way to translate human expertise into test scenarios
- Some of the technical ideas, such as autoregressive graph generation, are new and interesting.

Weaknesses:
- The comparisons provided in the paper are weak, and not representative of the state-of-the-art. (see review Gk4u)
- While some reviewers found the paper to be well-written, others identified issues with the clarity of the writing, especially around the assumptions made
- Several design choices are not motivated in the text

----

The authors provided a solid response that addressed some of the reviewer's concerns. There are some concerns that remain, e.g. around the task set-up from review Gk4u and as mentioned in the updated review pK7P, which I encourage the authors to improve for the revised version. Nonetheless, the paper makes an interesting contribution to CoRL.

**Best Paper Nomination:**

No

---

> ### Author Response · Authors · 2022-08-28
> **Response to meta review**
>
> We thank the Area Chair for summarizing the pre-rebuttal reviews. We highlight our revision as follows.
>
> 1. For the weak comparison issue, we add the following experiments: (i) comparison with state-of-the-art baselines, and (ii) adversarial training with generated scenarios. The results show that our methods could still achieve higher collision rates compared to the baselines under three different traffic scenario generation problems, and the usage of generated risk scenarios could significantly improve the generalization capability over different RL agents.
>
> 2. We clarify some important settings in the revised version, e.g. the definition of behavior graph, as well as the definition of node and edge attributes. To help understand how CausalAF functions during scenario generation, we also demonstrate the generated scenarios and provide qualitative results of the diversity of scenarios.
>
> 3. We further polish the description of motivation in this paper, including why we need the safety-critical scenarios, why we use normalizing flow, how to use generated scenarios, and how to extend the CausalAF to other robotics applications. We hope the quantitative and qualitative analysis in the paper is more convincing to all the reviewers.
>
> *Since we are heading to the deadline of the rebuttal phase, we hope our response has addressed the concerns of all reviewers. We encourage the reviewers to engage in the discussion if they have further questions.*